# The Prevalence of *Salmonella* spp. in Two Arctic Fox (*Alopex lagopus*) Farms in Poland

**DOI:** 10.3390/ani10091688

**Published:** 2020-09-18

**Authors:** Jan Siemionek, Konrad Przywara, Anna Szczerba-Turek

**Affiliations:** Department of Epizootiology, Faculty of Veterinary Medicine, University of Warmia and Mazury in Olsztyn, Oczapowskiego 13, 10-718 Olsztyn, Poland; jan.siemionek@uwm.edu.pl (J.S.); k.przywara@o2.pl (K.P.)

**Keywords:** *Salmonella* spp., epidemiology, one health approach, arctic foxes, Poland

## Abstract

**Simple Summary:**

*Salmonella enterica* subsp. *enterica* derived from poultry meat is the primary cause of *Salmonella* infection in humans and the second most ubiquitous zoonosis in the European Union after campylobacteriosis. Wildlife animals and livestock can be a reservoir of *Salmonella* spp., and they can contribute to the persistence of bacteria in the environment. *Salmonella* spp. pathogens can also be a source of widespread infections in fur-bearing animals, such as foxes *(Vulpes vulpes)* and mink (*Neovison vision*). This study analysed the prevalence of *Salmonella* spp. in two Arctic fox (*Alopex lagopus*) farms and the correlations between animals that tested positive for *Salmonella* spp and breeding results. *Salmonella* Heidelberg, *S.* Saintpaul, and *S.* Reading were isolated. All three serotypes are typically isolated from commercial poultry flocks. In this study, *Salmonella* spp. increased the risk of female infertility, but further research is needed to confirm the results. This is the first report on the prevalence of *S*. Heidelberg, *S.* Saintpaul, and *S.* Reading in an Arctic fox (*Alopex lagopus*) population.

**Abstract:**

The objective of the study was to determine the occurrence of *Salmonella* spp. infections in two Arctic fox (*Alopex lagopus*) farms in Poland, and to analyse the correlations between animals that tested positive for *Salmonella* spp and breeding results. Faecal samples were taken from 1094 clinically healthy blue foxes from the basic stock of farms A and B. *Salmonella* spp. were detected in 18.06% (56/310) of the samples collected in farm A and in 15.94% (125/784) of the samples collected in farm B. All isolated strains belonged to *S*. *enterica* subsp. *enterica* serotypes *Salmonella* Saintpaul (*S.* Saintpaul), *Salmonella* Reading (*S.* Reading), and *Salmonella* Heidelberg (*S*. Heidelberg). All three serotypes are typically isolated from commercial poultry flocks. *Salmonella* spp. infections significantly increased the risk of female infertility, but further research is needed to confirm the results. This is the first report on the prevalence of *S.* Heidelberg, *S.* Saintpaul, and *S*. Reading in faecal samples collected from Arctic fox (*Alopex lagopus*) farms in Poland.

## 1. Introduction

*Salmonella enterica* subsp. *enterica* derived from poultry meat is the primary cause of *Salmonella* infection in humans worldwide [1]. In 2018, 91,857 confirmed cases of *Salmonella* species (spp.) infections were registered in the European Union (EU), making salmonellosis the second most frequently reported zoonosis in the EU after campylobacteriosis (*N* = 246,571) [2]. More than 2500 serotypes have been identified to date, but most human infections are caused by only several serotypes [3]. In humans, *Salmonella* spp. pathogens are transmitted through food of animal origin, such as eggs, chicken, pork, or beef [4]. *Salmonella* spp. can be transmitted between animal species and between animals and humans, including by vectors [5,6,7]. *Salmonella* spp. are excreted with the faeces of healthy animals for long periods of time and can be isolated at any stage of the food chain [8,9]. *Salmonella* spp. are also one of the most common sources of infectious outbreaks in fur-bearing animals such as mink (*Neovison vision*) and foxes *(Vulpes vulpes)* [10]. The data on the epidemiological distribution of *Salmonella* spp. in mink and foxes are very limited. *Salmonella* spp. pathogens are transmitted when farmed animals are fed the meat of infected animals [11]. Feed of animal origin should constitute the bulk of fox diets (70–85%). The quality and composition of feed are not always strictly monitored. Feeds of low quality can introduce undesirable ingredients into the food chain. Fox feeds are composed mainly of fish, fish offal, poultry, poultry offal, abattoir offal, and cereals with mineral and vitamin supplements. These feeds are the main sources of infection in carnivorous fur-bearing animals [11,12]. *Salmonella* spp. infections lead to disorders of the digestive system, including abdominal and intestinal problems. Typical symptoms include diarrhoea, vomiting, and fever, but *Salmonella* spp. can also cause miscarriage and foetal death [10]. A high mortality rate was reported in infected newborns in the first days of life.

The objective of this study was to determine the occurrence of *Salmonella* spp. infections in two Arctic fox (*Alopex lagopus*) farms in Poland and to analyse the correlations between animals that tested positive for *Salmonella* spp. and breeding results.

## 2. Materials and Methods

### 2.1. Sample Collection

The study was performed on 1094 fresh faecal samples obtained from the basic stock of clinically healthy and variously aged male and female Arctic foxes (*Alopex lagopus*). The animals were bred in two farms (A and B) in the Polish region of Warmia and Mazury. The farms were selected by a veterinarian specialising in diseases of fur-bearing animals for convenience sampling. Arctic foxes were housed in individual cages according to the provisions of the Regulation of the Minister of Agriculture and Rural Development of 28 June 2010 [13]. All adult animals were tested only once. Between October 2014 and May 2015, a total of 310 of fresh faecal samples were collected underneath the cages occupied by 250 females and 60 males in farm A, and a total of 784 fresh faecal samples were collected underneath the cages occupied by 624 females and 160 males in farm B. Faecal samples were placed in sterile plastic tubes and transported in a refrigerator to the laboratory within 8 h.

### 2.2. Bacteriological Examination

Faecal samples of approximately 25 g each were initially incubated in 225 mL of Buffered Peptone Water (BPW) for 20–24 h at 37 °C under aseptic conditions. *Salmonella* spp. were isolated according to Polish Standards [14]. Five typical colonies were selected from each culture for further identification. Bacterial species were confirmed using the API20E biochemical test kit (bioMerieux, Marcy l’Etoile, France). All strains were stored at −80 °C in Brain Heart Infusion broth (Difco, BD, Franklin Lakes, NJ, USA) supplemented with 15% glycerine (Merck KGaA, Darmstadt, Germany).

### 2.3. Serotyping

Serotyping was performed by agglutination using commercial sera for O and H antigens (Difco, BD, Sparks, MD), according to Kauffmann–White classification [3].

### 2.4. Epidemiological and Statistical Analyses

The Clopper–Pearson ‘exact’ method based on the beta distribution at a significance level of α = 0.05 and 95% confidence interval was applied in basic statistical analyses. Data were processed statistically with EpiTools open-source epidemiological calculators [15]. The correlations between the number of females that had given birth/ females that had miscarried/ infertile females vs. the number of live-born/weaned cubs of Salmonella-positive and negative females were determined in Fisher’s exact test. Relative risk and attributable risk were calculated in cohort observational studies at a significance level of α = 0.05 and 95% confidence interval using the Working in Epidemiology (WinEpi) open-source programme. The reproduction rate and the case fatality rate (CFR) in cubs were calculated based on the adopted breeding indices for fur-bearing animals.

## 3. Results

In farm A, 14.4% of the analysed females (36/250, 95% CI = 10.29–19.37) and 33.33% of the males (20/60, 95% CI = 21.69–46.69) tested positive for Salmonella spp. All strains belonged to Salmonella enterica subsp. enterica serotype Heidelberg (S. Heidelberg) (n=56). In the group of Salmonella-positive females, 55.56% had given birth (20/36, 95% CI = 38.10–72.06), 16.67% had miscarried (6/36, 95% CI = 6.37–32.81), and 27.78% were infertile (10/36, 95% CI = 14.20–45.19). In the group of Salmonella-negative females, 70.56% had given birth (151/214, 95% CI = 63.96–76.58), 14.49% had miscarried (31/214 95% CI = 10.06–19.93), and 14.95% were infertile (32/214, 95% CI = 10.46–20.45). The results of bacteriological and serological analyses are shown in Table 1.

In farm B, 16.99% of the evaluated females (106/624, 95% CI = 14.12–20.12) and 11.88% of the males (19/160, 95% CI = 7.30–17.92) tested positive for Salmonella spp. Salmonella enterica subsp. enterica serotype Saintpaul (S. Saintpaul) was identified in 66.04% of the females (70/106, 95% CI = 56.20–74.96) and 68.42% of the males (13/19, 95% CI = 43.45–87.42). Salmonella enterica subsp. enterica serotype Reading (S. Reading) was identified in 33.96% of the females (36/106, 95% CI = 25.04–43.80) and 31.58% of the males (6/19, 95% CI = 12.58–56.55). The results are shown in Table 1. In the group of Salmonella-positive females, 82.08% had given birth (87/106, 95% CI = 73.43–88.85), 10.38% had miscarried (11/106, 95% CI = 5.30–17.81), and 7.55% were infertile (8/106, 95% CI = 3.31–14.33). In the group of Salmonella-negative females, 86.68% had given birth (449/518, 95% CI = 83.45–89.49), 10.23% had miscarried (53/518, 95% CI = 7.76–13.17), and 3.09% were infertile (16/518, 95% CI = 1.78–4.97). The correlations between the number of Salmonella-positive and negative females that had given birth/had miscarried/were infertile were not significant (*p*-values in the two-tailed test: 0.08, 0.79, 0.08 in farm A, and 0.22, 1.00, 0.04 in farm B, respectively).

The calculated risk estimates for *Salmonella*-positive females that had given birth/had miscarried/were infertile differed between farms. No significant differences were observed among *Salmonella*-positive females that had given birth/had miscarried, but in the group of infertile females, the relative risk was determined at 1.85 (95% CI = 1.00–3.43) in farm A and 2.44 (95% CI = 1.07–5.56) in farm B, respectively. The risk of infertility in *Salmonella*-positive females was determined at 12.82% in farm A and 4.46% in farm B, respectively.

The number of live-born/weaned cubs delivered by *Salmonella*-positive and negative females was determined at 3.20/3.15 and 5.23/4.85 per mother in farm A, and at 9.38/7.06 and 9.57/7.03 per mother in farm B, respectively. The results are collated in Table 1. The case fatality rate (CFR) among the offspring of *Salmonella*-positive and negative females was determined at 1.59% and 7.34% in farm A, and at 24.63% and 26.49% in farm B, respectively. The correlations between the number of live-born/weaned cubs delivered by *Salmonella*-positive and negative females were not statistically significant. Cub mortality was 23.07% higher in farm B than in farm A.

## 4. Discussion

The One Health approach was developed in the early 2000s with the goal of recognising the continuous, cumulative and globally interdependent interactions between causes and effects within ecosystems and their human and animal populations. This approach can be applied to food security, economic sustainability, and animal welfare [16,17]. Free-living red foxes (*Vulpes vulpes*) can be a reservoir of *Salmonella* spp. In an Italian study, *Salmonella* spp. were identified in 5.7% of the tested samples, in Austria at 2.1%, and in Norway at 6.5%, respectively [18,19,20]. In Poland, the prevalence of *Salmonella* spp. in free-living foxes *(Vulpes vulpes)* was determined at 3.15% [21]. The prevalence of *Salmonella* spp. in farmed foxes reached 4.2% [22]. These studies indicate that free-living red foxes (*Vulpes vulpes*) harbour numerous *Salmonella* spp. serotypes which are also responsible for infections in Europe [23]. In the present study, the prevalence of *Salmonella* spp. in Arctic foxes (*Alopex lagopus*) was determined at 18.06% (56/310) in farm A, and 15.94% (125/784) in farm B. All isolated serotypes (*S.* Saintpaul, *S.* Reading, and *S*. Heidelberg) belonged to *S*. *enterica* subsp. *enterica*. The identified serotypes are commonly encountered in commercial poultry farms which are the primary reservoirs of all *Salmonella* spp. serotypes in different countries [24,25] and may trigger human infections around the world [25,26,27]. The presence of these poultry pathogens in foxes results from feeding these animals with poultry slaughter waste, mainly from turkey processing. In 1999 and 2009, *S*. Saintpaul, *S*. Senftenberg, *S*. Anatum, *S.* Heidelberg, *S*. Hadar, *S*. Typhimurium, *S.* Infantis, *S*. Enteritidis, and *S*. Agona were isolated from chicken and turkey carcasses in north-eastern Poland [12,28]. In 2014 and 2016, *S*. Enteritidis, *S*. Typhimurium, *S*. Anatum, *S.* Kentucky, *S*. Infantis, and *S*. Mbandaka were isolated from broiler chicken flocks in the same region [29]. Warmia and Mazury is a leading producer of turkeys in Poland, and it is characterised by high availability of poultry slaughter waste, including hard waste such as heads and legs, or soft waste such as meat and bone meal. These nutritious and energy-dense feed components are often acquired and stored in poor sanitary conditions, which can lead to the contamination of entire feed lots, mainly with *Salmonella* rods. Hygiene standards in hatcheries and poultry processing plants have to be improved to prevent the circulation of *Salmonella* spp. serotypes in poultry feed, animals, and the environment [30].

Correlations between animals that tested positive for *Salmonella* spp and the breeding results of Arctic foxes (*Alopex lagopus*) were also determined in the study. The only statistically significant result was that in both farms, *Salmonella*-positive females were at higher risk of infertility. Reproductive problems have a complex aetiology in Arctic foxes, and numerous factors that affect fertility need to be controlled, including in statistical analyses. In the examined farms, CFR values were not correlated with the number of *Salmonella*-positive or negative females. The risk of cub mortality was around 23.07% higher in the larger farm (B), but further research is needed to confirm these results.

## 5. Conclusions

In the present study, the prevalence of *Salmonella* spp. in Arctic foxes (*Alopex lagopus*) in the examined farms was estimated between 15.94% and 18.06%. All isolated strains were identified as serotypes *S.* Saintpaul, *S.* Reading, and *S*. Heidelberg of *S*. *enterica* subsp. *enterica* which are potentially pathogenic for humans. The risk of infertility was significantly higher in the population of *Salmonella*-positive females at 4.46–12.82%. This is the first report on the prevalence of *S.* Heidelberg, *S.* Saintpaul, and *S*. Reading in faecal samples collected from Arctic fox (*Alopex lagopus*) farms in Poland.

## Figures and Tables

**Table 1 animals-10-01688-t001:** Prevalence of *Salmonella* spp. and reproductive success in Arctic foxes (*Alopex lagopus*).

Reproductive SuccessTotal	Farm AFemales250	Farm AMales60	Farm BFemales624	Farm BMales160
	Salmonella-Negative	Salmonella-Positive	Salmonella-Negative	Salmonella-Positive	Salmonella-Negative	Salmonella-Positive	Salmonella-Negative	Salmonella-Positive
*Total*	214	36	40	20	518	106	141	19
*No. of* *females* *that gave birth*	151	20	--	--	449	87	--	--
*No. of females that miscarried*	31	6	--	--	53	11	--	--
*No. of infertile females*	32	10	--	--	16	8	--	--
*No. of live-born cubs/No. of females that* *gave birth*	790/151 = 5.23	64/20 = 3.2	--	--	4299/449 = 9.57	816/87 = 9.38	--	--
*No. of weaned puppies/No. of females* *that gave birth*	732/151 = 4.85	63/20 = 3.15	--	--	3160/449 = 7.03	615/87 = 7.06	--	--
*Case fatality rate (CFR)*	7.34	1.56	--	--	26.49	24.63		
*S. enterica subsp. enterica* serotype								
Saintpaul						70		13
Reading						36		6
Heidelberg		36		20

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
