# Peer review of "The Prevalence of Salmonella spp. in Two Arctic Fox (Alopex lagopus) Farms in Poland"

_animals, 2020, doi:10.3390/ani10091688_

Round 1

Reviewer 1 Report

In the manuscript entitled "Prevalence of Salmonella spp in Blue Fox Farms (Alopex lagopus) in Poland” the authors examined the frequency of Salmonella presence in faeces samples taken from two farms of Blue foxes, and also identified Salmonella serovars isolated. Besides, they study the relationship between Salmonella presence and reproductive problems. The subject of the manuscript is interesting and essential to increase Salmonella epidemiology knowledge. However, I have some important comments before I can recommend the manuscript for publication in Animals.

General comments:

English has to be reviewed again by an English reviewer before uploaded to the platform again.

The objective of the work is not adequately described. The authors mix different concepts: one-health, fox reproductive problems, contaminated feeding (poultry products), and related serovars. Inline, it is necessary to rewrite the introduction and discussion to focus the real objective of their study.

The data must always be expressed with the same number of decimals throughout the manuscript—preferably 1 single decimal.

Specific comments:

Title: The authors of the manuscript didn’t study the prevalence of Salmonella in Blue Fox Farms in Poland; they study the percentage of Salmonella in two fox farms in Poland. For this reason, “prevalence” is not the proper epidemiological concept to include in the title and through the manuscript.

Introduction

Line 44: change enerica by enterica

Line 54: Thru?

Lines 59-62 (“…but in contrast to the abundance….”): confusing paragraph. No sense. Please, rewrite for better understanding.

Lines 65-67 (“Microbiological contamination….”): confusing paragraph. No sense. Please, rewrite for better understanding.

Lines 71: What’s a general public disease means? Public health disease?

Lines 78-80: the objective is different in the Simple summary, abstract, and manuscript. The authors of this manuscript collected faeces samples from two fox farms, isolated and serotyped Salmonella, and related the presence or not of Salmonella with reproductive problems. Inline, please rewrite the objective. 

Material and Methods

Lines 82-88 (sample collection): Please include how many faeces (g.) have been collected for each animal at farm level; they were or not fresh faeces. It was a pool from different faeces; a rectal swab collected directly from the fox or the floor. It is important information to know if the samples are or not independent and enough for Salmonella isolation.

Lines 96-98 (Serotyping): 5 colonies by animal/sample were identified, how many colonies by animal were serotyped?

Lines 99-108: Rewrite the statistical analysis section. It is important to describe the analysis in deep, according to the objective of the study. 

Lines 110-111: include the values used in the analysis. 

Results

Lines 113-114: Remove the sentence “In the studied population of 310 blue foxes (Alopex lagopus) on farm A, females were 250 (80.65%, 95% CI = 75.8 – 84.89) and males 60 (19.35%, 95% CI = 15.11 – 24.2)”. This data is included in the material and methods; it is redundant to include the distribution of the population again in results.

Lines 116-117: Remove the sentence “and 214 negatives for Salmonella spp. (85.6%, 95% CI = 80.63 – 89.71). It is redundant to include the negative population if you have previously described the positive population.

Lines 113-120: I strongly suggest rewriting the paragraph as follows: In the studied population from farm A, 14.4% of females were positive for Salmonella infection (36/250, 95%, CI=10.29-19.37). All strains were S. Heidelberg serovar (n=X)...

Lines 120-124: Include in material and methods that data relating reproductive problems have been collected. Also, include in the statistical analysis section that you have divided for the statistical analysis of the animals in two main groups: Salmonella positive and negative animals (carriers or not).

Lines 124-126: remove how the statistical analysis has been done from the results section; this sentence has to be included in the material and methods section (statistical analysis).

Lines 129-145: Results section from farm B have to be rewritten as indicated previously for farm A.

Conclusions

Conclusions have to respond to your objectives; please rewrite them.

Simple summary and abstract

According to previous comments and suggestions, the “Simple summary and abstract” has to be rewritten for better understanding and adapted to 200 words according to the guidelines. 

Author Response

18.08.2020

The Authors would like to thank the Editor and Reviewers for a thorough perusal of our manuscript, and for the valuable comments and suggestions that helped us improve the quality of the paper. Below are point-by-point responses to the comments made by Reviewer 1. All content-related changes introduced to the text are highlighted in yellow, and linguistic corrections are marked in red.

Reviewer #1

General comments:

Comment 1:

English has to be reviewed again by an English reviewer before uploaded to the platform again.

Author’s response:

The entire manuscript has been revised by a professional translation service, and it has also been spell-checked and grammar-checked by a native English speaker. The linguistic corrections are marked in red in the revised manuscript.

Comment 2:

The objective of the work is not adequately described. The authors mix different concepts: one-health, fox reproductive problems, contaminated feeding (poultry products), and related serovars. Inline, it is necessary to rewrite the introduction and discussion to focus the real objective of their study.

Author’s response:

The relevant corrections have been made.

Comment 3:

The data must always be expressed with the same number of decimals throughout the manuscript—preferably 1 single decimal.

Author’s response:

The relevant corrections have been made: in the revised manuscript, the data are expressed accurate to two decimal places.

Specific comments:

Comment 1:

Title: The authors of the manuscript didn’t study the prevalence of Salmonella in Blue Fox Farms in Poland; they study the percentage of Salmonella in two fox farms in Poland. For this reason, “prevalence” is not the proper epidemiological concept to include in the title and through the manuscript.

Author’s response:

In the manuscript, the term “prevalence” is not used in the epidemiological sense, but rather as “dispersion” or “spread”.

Comment 2:

Line 44: change enerica by enterica

Author’s response:

The relevant correction has been made.

Comment 3:

Line 54: Thru?

Author’s response:

“thru” has been replaced with the correct word “through”

Comment 4:

Lines 59-62 (“…but in contrast to the abundance….”): confusing paragraph. No sense. Please, rewrite for better understanding.

Author’s response:

The sentence has been rewritten in line with the Reviewer’s suggestion.

Comment 5:

Lines 65-67 (“Microbiological contamination….”): confusing paragraph. No sense. Please, rewrite for better understanding.

Author’s response:

The above sentence has been removed.

Comment 6:

Lines 71: What’s a general public disease means? Public health disease?

Author’s response:

The relevant correction has been made.

Comment 7:

Lines 78-80: the objective is different in the Simple summary, abstract, and manuscript. The authors of this manuscript collected faeces samples from two fox farms, isolated and serotyped Salmonella, and related the presence or not of Salmonella with reproductive problems. Inline, please rewrite the objective. 

Author’s response:

The objective of the study has been rewritten.

Comment 8:

Lines 82-88 (sample collection): Please include how many faeces (g.) have been collected for each animal at farm level; they were or not fresh faeces. It was a pool from different faeces; a rectal swab collected directly from the fox or the floor. It is important information to know if the samples are or not independent and enough for Salmonella isolation.

Author’s response:

We collected only fresh faeces underneath the cages, from one animal, and faecal samples of approximately 25 g each were analysed. In basic stock, one animal is kept in one cage, therefore the faeces was not a pool.

Comment 9:

Lines 96-98 (Serotyping): 5 colonies by animal/sample were identified, how many colonies by animal were serotyped?

Author’s response:

Fresh faecal samples were obtained underneath the cages (with one animal per cage) and after bacteriological examinations, 5 typical colonies were selected from each culture for further identification.

Comment 10:

Lines 99-108: Rewrite the statistical analysis section. It is important to describe the analysis in deep, according to the objective of the study. 

Author’s response:

The relevant corrections have been made

Comment 11:

Lines 110-111: include the values used in the analysis. 

Author’s response:

The values were included in the last paragraph of the Results section.

Comment 12:

Lines 113-114: Remove the sentence “In the studied population of 310 blue foxes (Alopex lagopus) on farm A, females were 250 (80.65%, 95% CI = 75.8 – 84.89) and males 60 (19.35%, 95% CI = 15.11 – 24.2)”. This data is included in the material and methods; it is redundant to include the distribution of the population again in results.

Author’s response:

The above sentence has been removed.

Comment 13:

Lines 116-117: Remove the sentence “and 214 negatives for Salmonella spp. (85.6%, 95% CI = 80.63 – 89.71). It is redundant to include the negative population if you have previously described the positive population.

Author’s response:

The relevant corrections have been made

Comment 14:

Lines 113-120: I strongly suggest rewriting the paragraph as follows: In the studied population from farm A, 14.4% of females were positive for Salmonella infection (36/250, 95%, CI=10.29-19.37). All strains were S. Heidelberg serovar (n=X)..

Author’s response:

The relevant corrections have been made

Comment 15:

Lines 120-124: Include in material and methods that data relating reproductive problems have been collected. Also, include in the statistical analysis section that you have divided for the statistical analysis of the animals in two main groups: Salmonella positive and negative animals (carriers or not).

Author’s response:

The relevant corrections have been made

Comment 16:

Lines 124-126: remove how the statistical analysis has been done from the results section; this sentence has to be included in the material and methods section (statistical analysis).

Author’s response:

The relevant corrections have been made

Comment 17:

Lines 129-145: Results section from farm B have to be rewritten as indicated previously for farm A.

Author’s response:

The relevant corrections have been made

Comment 18:

Conclusions have to respond to your objectives; please rewrite them.

Author’s response:

The relevant corrections have been made

Comment 19:

According to previous comments and suggestions, the “Simple summary and abstract” has to be rewritten for better understanding and adapted to 200 words according to the guidelines. 

Author’s response:

The relevant corrections have been made

Reviewer 2 Report

Prevalence of Salmonella spp. In Blue Fox Farms in Poland

Study: This is a Salmonella prevalence study with sampling of foxes in two Blue fox farms in Poland. One faecal sample from all animals on the farms were collected within a six month period. Samples were cultured for Salmonella spp. and positive cultures were serotyped.

Data on sampled individuals was collected and associations between salmonella status and fertility factors in females (whelping, aborting and non-pregnant females) was checked for.

Comments

According to the authors, this study is performed in an animal population (farmed foxes) with little previous knowledge on salmonella occurence, which makes it interesting and relevant for publication. However, the description of how the sampling was performed needs to be improved. Please clarify how the two sampled farms where selected, the number of animals present in each farm, how animals are kept and how much contact there is between animals, the number of sampled animals in each farm (my guess is that all animals in the farms were sampled, but it is not clearly described), how animals within the farms were selected for sampling (in case all not were sampled), the number of times each animal was sampled, and how samples were taken (rectal?, from the cage floor? Other?). The sampling took place during a six month period which is an extended time period for a prevalence study, where each animal only sampled once?

The study is not well designed to evaluate the effect of salmonella infection on fertility. The reason is that both farms have a high prevalence of salmonella and when salmonella is present in a farm, it normally circulates between the individual animals with individual animals only being infected with salmonella for a couple of days or weeks. Therefore, when salmonella not is isolated from an animal, salmonella may very well be isolated from that same animal if sampled again the next week or month. The authors use the term carriers for test positive animals. This is a term usually used for chronic carriers of salmonella infection and in most species this predominately occurs in species adapted serotypes and in a small proportion of infected animals. If each animal in this study only was tested once, there is no support for the salmonella positive foxes to be chronic carriers.

This means that in order to study the effect on fertility, comparison would preferably be done comparing a population with salmonella to a population without salmonella infection, for example comparing fertility in farms with salmonella infection to fertility in farms without salmonella infection. However, also that is difficult, as there are many factors that may have effect on fertility, and these factors need to be controlled for in the statistical analyses. In addition, different serotypes of salmonella may have different effects on the fertility. Please either remove this part of the study, or include these considerations in the discussion.

The discussion includes results from other studies in wild foxes. It would be more interesting if the discussion would include results and references from salmonella studies in Polish poultry, as the feed to the foxes include poultry slaughter waste and this is the likely infection route suggested by the authors. It would also be nice if the authors could include considerations on potential feed treatments to reduce the risk of salmonella in the feed.

It is also interesting that different salmonella serotypes were isolated from the two different farms. Do salmonella serotypes in poultry differ between regions in Poland? Maybe this could also be addressed in the discussion. Was the feed in the farms sampled?

The authors suggest that spread of manure from fox farms possesses a risk to humans and therefore the manure should be composted before spread. This needs to be supported with references.

Author Response

18.08.2020.

The Authors would like to thank the Editor and Reviewers for a thorough perusal of our manuscript, and for the valuable comments and suggestions that helped us improve the quality of the paper.  Below are point-by-point responses to the comments made by Reviewer 2. All content-related changes introduced to the text are highlighted in yellow, and linguistic corrections are marked in red.

Reviewer #2

Comment 1:

According to the authors, this study is performed in an animal population (farmed foxes) with little previous knowledge on salmonella occurence, which makes it interesting and relevant for publication. However, the description of how the sampling was performed needs to be improved. Please clarify how the two sampled farms where selected, the number of animals present in each farm, how animals are kept and how much contact there is between animals, the number of sampled animals in each farm (my guess is that all animals in the farms were sampled, but it is not clearly described), how animals within the farms were selected for sampling (in case all not were sampled), the number of times each animal was sampled, and how samples were taken (rectal?, from the cage floor? Other?). The sampling took place during a six month period which is an extended time period for a prevalence study, where each animal only sampled once?

Author’s response:

The relevant corrections have been made, the missing information has been provided, and the entire subsection 2.1. has been rewritten.

Comment 2:

The study is not well designed to evaluate the effect of salmonella infection on fertility. The reason is that both farms have a high prevalence of salmonella and when salmonella is present in a farm, it normally circulates between the individual animals with individual animals only being infected with salmonella for a couple of days or weeks. Therefore, when salmonella not is isolated from an animal, salmonella may very well be isolated from that same animal if sampled again the next week or month. The authors use the term carriers for test positive animals. This is a term usually used for chronic carriers of salmonella infection and in most species this predominately occurs in species adapted serotypes and in a small proportion of infected animals. If each animal in this study only was tested once, there is no support for the salmonella positive foxes to be chronic carriers.

Author’s response:

The relevant corrections have been made.

Comment 3:

This means that in order to study the effect on fertility, comparison would preferably be done comparing a population with salmonella to a population without salmonella infection, for example comparing fertility in farms with salmonella infection to fertility in farms without salmonella infection. However, also that is difficult, as there are many factors that may have effect on fertility, and these factors need to be controlled for in the statistical analyses. In addition, different serotypes of salmonella may have different effects on the fertility. Please either remove this part of the study, or include these considerations in the discussion.

Author’s response:

We found that the risk of sterility was significantly higher in Salmonella-positive females. We are aware that this is insufficient to formulate far-reaching conclusions. However, the number of studies investigating this problem is limited, and further research is needed.

Comment 4:

The discussion includes results from other studies in wild foxes. It would be more interesting if the discussion would include results and references from salmonella studies in Polish poultry, as the feed to the foxes include poultry slaughter waste and this is the likely infection route suggested by the authors. It would also be nice if the authors could include considerations on potential feed treatments to reduce the risk of salmonella in the feed.

Author’s response:

The relevant corrections have been made

Comment 5:

It is also interesting that different salmonella serotypes were isolated from the two different farms. Do salmonella serotypes in poultry differ between regions in Poland? Maybe this could also be addressed in the discussion. Was the feed in the farms sampled?

Author’s response:

No, the feed in the farms was not sampled.

Comment 6:

The authors suggest that spread of manure from fox farms possesses a risk to humans and therefore the manure should be composted before spread. This needs to be supported with references.

Author’s response:

The above information has been removed because the information regarding the contamination of poultry feed seems to be more important.

Reviewer 3 Report

The manuscript "Prevalence of Salmonella spp. in blue fox farms (Alopex lagopus) in Poland" is dedicated to the isolation of three Salmonella serovars from Arctic foxes in two farms in Poland. The authors claim that this is the first description of the isolation of Salmonella from Arctic foxes in Poland.

Since the manuscript was written by non-English-speaking authors, I would highly recommend significant professional editing of the English. The translation into English is extremely sloppy and difficult to read. Some words are used incorrectly and it is not clear what the authors wanted to say. Unfortunately, the authors found almost no statistically significant relationships. In this regard, at the discretion of the authors, I would recommend shortening the article and making it a Short Communication. I would like to know how the isolated strains of the three Salmonella serovars are related to each other using at least one of the molecular genetic methods (plasmid profile analysis, PFGE, MLST, MLVA, WGS, etc.). Hopefully, this will be done in the next publication.

Below are the minor issues of the manuscript:
1) In the title of the manuscript and in the text, instead of a blue fox, I would recommend using the more traditional animal name Arctic fox, white fox, polar fox, or snow fox.
2) Line 2: In the title of the manuscript, the Latin name should be placed after the name of the animal, and not after the word "farm".
3) Line 15: The species name is written with a small letter (Vulpes vulpes) (also on lines 58, 194, 200, 205).
4) Line 22: You cannot talk about the pathogenicity of these specific isolated Salmonella isolates, since this is not shown in the text of the manuscript.
5) Line 31 etc.: Choose one of the terms "serovar" or "serotype" and use it throughout the manuscript. Both are now used in the text.
6) Line 36 etc.: In the text of the manuscript: The wrong abbreviation CSF (cerebrospinal fluid) is used instead of CFR (case-control ration).
7) Line 45: Choose one: salmonellosis or Salmonella infection.
8) Line 49-50: The sentence is unfinished.
9) Line 51: There is nothing about the transmission from humans to animals in reference [7].
10) Line 91: What is BTL?
11) Line 116 etc.: Subsp. and serovar should not be written in italic!
12) Lines 187-193: Don't repeat the same thing twice. Everyone has already understood that S. is Salmonella.
13) Lines 208-209: S. Reading and S. Heidelberg have already been given earlier (lines 116, 132-133).
14) Line 209: The total percentage must be 100%, not 200%.
15) Line 218: A reference why the term "pulp" is used in this form?
16) Line 226: "Data", not "date".
17) Line 228: CFR has already been explained earlier.
18) Lines 238-239: Better to use the correct animal names: red fox instead of a common fox; Arctic fox, white fox, polar fox, or snow fox instead of a blue fox, Neovison vison instead of Mustela vison, raccoon dog instead of yenot, coypu instead of nutrient.
19) References must be reduced from 38 to 30, as it is written in the Instructions for Authors.

Author Response

18.08.2020.

The Authors would like to thank the Editor and Reviewers for a thorough perusal of our manuscript, and for the valuable comments and suggestions that helped us improve the quality of the paper. Below are point-by-point responses to the comments made by Reviewer 3. All content-related changes introduced to the text are highlighted in yellow, and linguistic corrections are marked in red.

Reviewer #3

General comment:

Since the manuscript was written by non-English-speaking authors, I would highly recommend significant professional editing of the English. The translation into English is extremely sloppy and difficult to read. Some words are used incorrectly and it is not clear what the authors wanted to say. Unfortunately, the authors found almost no statistically significant relationships. In this regard, at the discretion of the authors, I would recommend shortening the article and making it a Short Communication. I would like to know how the isolated strains of the three Salmonella serovars are related to each other using at least one of the molecular genetic methods (plasmid profile analysis, PFGE, MLST, MLVA, WGS, etc.). Hopefully, this will be done in the next publication.

Author’s response:

The entire manuscript has been revised by a professional translation service, and it has also been spell-checked and grammar-checked by a native English speaker. The linguistic corrections are marked in red in the revised manuscript

As recommended by the Reviewer, the article has been shortened and presented as a Short Communication.

Currently, molecular methods were not used to check similarity of the isolated strains, but it will be done in our next study.

Comment 1:
In the title of the manuscript and in the text, instead of a blue fox, I would recommend using the more traditional animal name Arctic fox, white fox, polar fox, or snow fox.

Author’s response:

The relevant corrections have been made

Comment 2:
Line 2: In the title of the manuscript, the Latin name should be placed after the name of the animal, and not after the word "farm".

Author’s response:

The relevant correction has been made

Comment 3:
Line 15: The species name is written with a small letter (Vulpes vulpes) (also on lines 58, 194, 200, 205).

Author’s response:

The relevant corrections have been made

Comment 4:
Line 22: You cannot talk about the pathogenicity of these specific isolated Salmonella isolates, since this is not shown in the text of the manuscript.

Author’s response:

The above sentence has been removed.

Comment 5:

Line 31 etc.: Choose one of the terms "serovar" or "serotype" and use it throughout the manuscript. Both are now used in the text.

Author’s response:

The relevant corrections have been made, we have decided on "serotype" 

Comment 6:

Line 36 etc.: In the text of the manuscript: The wrong abbreviation CSF (cerebrospinal fluid) is used instead of CFR (case-control ration).

Author’s response:

The relevant corrections have been made

Comment 7:
Line 45: Choose one: salmonellosis or Salmonella infection.

Author’s response:

We prefer “Salmonella infection”, and the relevant corrections have been made

Comment 8:
Line 49-50: The sentence is unfinished.

Author’s response:

“Although” has been removed from this sentence.

Comment 9:
Line 51: There is nothing about the transmission from humans to animals in reference [7].

Author’s response:

The phrase “or from humans to animals” has been removed from this sentence.

Comment 10:
Line 91: What is BTL?

Author’s response:

The abbreviation “BTL” has been removed, we apologise for this mistake.

Comment 11:
Line 116 etc.: Subsp. and serovar should not be written in italic!

Author’s response:

The relevant corrections have been made

Comment 12:
Lines 187-193: Don't repeat the same thing twice. Everyone has already understood that S. is Salmonella.

Author’s response:

The relevant corrections have been made

Comment 13:
Lines 208-209: S. Reading and S. Heidelberg have already been given earlier (lines 116, 132-133).

Author’s response:

The relevant corrections have been made

Comment 14:
14) Line 209: The total percentage must be 100%, not 200%.

Author’s response:

The relevant corrections have been made

Comment 15:
15) Line 218: A reference why the term "pulp" is used in this form?

Author’s response:

It has been removed.

Comment 16:
Line 226: "Data", not "date".

Author’s response:

The relevant corrections have been made

Comment 17:
Line 228: CFR has already been explained earlier.

Author’s response:

The relevant corrections have been made

Comment 18:
Lines 238-239: Better to use the correct animal names: red fox instead of a common fox; Arctic fox, white fox, polar fox, or snow fox instead of a blue fox, Neovison vison instead of Mustela vison, raccoon dog instead of yenot, coypu instead of nutrient.

Author’s response:

The relevant corrections have been made, and the above passage has been removed from the revised manuscript.

Comment 19:
References must be reduced from 38 to 30, as it is written in the Instructions for Authors.

Author’s response:

The number of References has been reduced to 30.

Round 2

Reviewer 1 Report

The authors of the study improved in deep the study and I recommend the publication in present form.

Author Response

Thank You so much,

All the best

Anna

Reviewer 2 Report

Title: I preferred the previous title, with the addition that it is a study in two farms.

The paper has been improved and there is now a comprehensive description on how the study was performed, but I still don´t find it ready for publication. See comments below.

Major comments;

In this study the authors have found an association between two factors, a salmonella positive test result and infertility in female foxes. It does not automatically mean that a salmonella positive test is a risk factor for infertility as no causality has been shown. Please, clarify this in throughout paper. Also, in my previous review I asked the authors to include a part in the discussion, on the design of the study and the problems in with the design of the study relating to studying risk factors. “The study is not well designed to evaluate the effect of salmonella infection on fertility. The reason is that both farms have a high prevalence of salmonella and when salmonella is present in a farm, it normally circulates between the individual animals with individual animals only being infected with salmonella for a couple of days or weeks. Therefore, when salmonella not is isolated from an animal, salmonella may very well be isolated from that same animal if sampled again the next week or month”.

I still would like to see a part in the discussion around this, as it could also explain that no other associations were found between test-positive animals and the fertility factors tested for. It would also be relevant for the authors to include information on when the salmonella samples were taken in relation to when abortions, successful mating, non-successful mating etc occurred. The authors also need to describe how they define “sterile” female.

On line 132 the authors write “An epidemiological analysis revealed the CRF was related to farm size.” The study only includes two farms, and therefore it is not possible to evaluate the relation between farm size and case fatality rate in different farms. Please remove this part.

The English has been improved, but still the are sentences that I cannot understand (for example line 41 and line 46-47). And in line 99 the authors write about a cohort study not described previously.

Also, the authors need to consider their choice of words. For example, the work linked is used in several places, when I think the authors mean that the same serotypes occur I different populations. Also, the authors write that there are salmonella serotypes in wild foxes “responsible for infections in humans”. To my knowledge, it has not been shown that humans are infected by wild foxes, and I think the authors mean that the same serotypes are found in the different populations. The authors also write that the severity of salmonella infections in farmed foxes has been studied, when the study concerns occurrence of salmonella in two farms. The word determined is used, when I think the authors mean testing for associations. Please check that the correct expressions are used throughout the paper.

In line 55 the authors write that “salmonella is most prevalent in farms where veterinary and sanitary regulations are not observerd”. Please include the reference for this.

Author Response

The Authors would like to thank the Editor and the Reviewers for a thorough perusal of our manuscript, and for the valuable comments and suggestions that helped us improve the quality of the paper. Our point-by-point responses to the comments made by Reviewer 2 in Round 2 of the revision process are presented below. All content-related changes in the text are highlighted in green, and linguistic corrections are marked in blue.

General comments:

Comment 1:

Title: I preferred the previous title, with the addition that it is a study in two farms.

Author’s response:

According to the Reviewer’s suggestion, the title has been changed to: “The Prevalence of Salmonella spp. in two Arctic Fox (Alopex lagopus) Farms in Poland”.

Major comments:

Comment 1 and Comment 2

In this study the authors have found an association between two factors, a salmonella positive test result and infertility in female foxes. It does not automatically mean that a salmonella positive test is a risk factor for infertility as no causality has been shown. Please, clarify this in throughout paper. Also, in my previous review I asked the authors to include a part in the discussion, on the design of the study and the problems in with the design of the study relating to studying risk factors. “The study is not well designed to evaluate the effect of salmonella infection on fertility. The reason is that both farms have a high prevalence of salmonella and when salmonella is present in a farm, it normally circulates between the individual animals with individual animals only being infected with salmonella for a couple of days or weeks. Therefore, when salmonella not is isolated from an animal, salmonella may very well be isolated from that same animal if sampled again the next week or month”.

I still would like to see a part in the discussion around this, as it could also explain that no other associations were found between test-positive animals and the fertility factors tested for. It would also be relevant for the authors to include information on when the salmonella samples were taken in relation to when abortions, successful mating, non-successful mating etc occurred. The authors also need to describe how they define “sterile” female.

Author’s response:

“In this study the authors have found an association between two factors, a salmonella positive test result and infertility in female foxes. It does not automatically mean that a salmonella positive test is a risk factor for infertility as no causality has been shown.” Only a correlation analysis was performed, and the results were statistically significant, but further and more detailed research is needed. The samples were collected only once from each animal, and they consisted of faeces, not vaginal or foreskin smears. The samples were examined for the presence of Salmonella spp. in farmed foxes. Farm owners reported breeding disorders, and they provided the researchers with data on abortions, successful and non-successful mating. Statistical and epidemiological methods were used to determine the presence of correlations between breeding disorders and infections caused by S. Saintpaul, S. Reading and S. Heidelberg. Significant correlations were found only with female infertility. The correlations between female infertility and positive results of Salmonella spp. tests should be analysed in greater depth, and vaginal smear samples should be collected at different time intervals. We agree with the Reviewer that a correlation between the presence of Salmonella spp. in vaginal smears and breeding results should be analyzed and determined, but farm owners did not consent to such tests in fear of further losses. The collection of faecal samples from underneath the cages is a stressful experience for the animals during the breeding period, and it can contribute to miscarriage. Two months before the present study, one of the co-authors, Professor Jan Siemionek, the national expert for diseases in fur-bearing animals, had performed a preliminary bacteriological analysis to identify Salmonella spp. carriers. The results were not published. In the preliminary analysis, faecal samples were collected once a week from clinically healthy female foxes over a period of six weeks. The analysis revealed the presence of S. Saintpaul (which is typically identified in poultry) in 4.1% of all faecal samples, and in 11.3% of the samples during the second analysis. Only three females tested positive in the second test that was conducted one week later. Based on these results, the authors concluded that a single test during the breeding period was sufficient. Miscarriages caused by Salmonella serotypes characteristic of poultry were not noted in Arctic foxes in Poland, but Salmonella infections had been previously reported by Yan et al. (1997) in female Arctic foxes and by Sato and Kuwamoro (1999) in the family Canidae. The terms “sterile/sterility” were replaced with “infertile/female infertility” throughout the manuscript. Infertile females were defined as females that did not present with clinical symptoms of pregnancy and did not produce offspring.

The authors are researchers and academic lecturers of the Department of Epizootiology, and they are aware that the Reviewer’s suggestions should be taken into account in epidemiological studies. However, the Reviewer’s remarks could not be considered in the current study, and they will be taken into account in further research on bacterial infections in farmed foxes. The present study may be incomplete, but it provides unique and valuable information, stressing the need for further research.

Comment 3:

On line 132 the authors write “An epidemiological analysis revealed the CRF was related to farm size.” The study only includes two farms, and therefore it is not possible to evaluate the relation between farm size and case fatality rate in different farms. Please remove this part.

Author’s response:

The following sentences and fragments have been removed:  

Lines 131-133: “An epidemiological analysis revealed that CFR was related to farm size. Relative risk was determined at 15.76 in farm B (95% CI =2.24 – 110.62), and the size of farm B was a risk factor for cub mortality.”

and

lines 174-175: “, but with farm size.”

Comment 4:

The English has been improved, but still the are sentences that I cannot understand (for example line 41 and line 46-47). And in line 99 the authors write about a cohort study not described previously.

Author’s response:

The manuscript has been edited by a professional translator.

Lines 92-94: “Relative risk and attributable risk were calculated in cohort observational studies at a significance level of a = 0.05 and 95% confidence interval using the Working in Epidemiology (WinEpi) open-source programme (http://www.winepi.net/uk/index.htm)”. The term “cohort” was used in the Materials and Methods section (2.4. Epidemiological and statistical analyses) to indicate which WinEpi tab should be opened to calculate RR or AR.

Comment 5:

Also, the authors need to consider their choice of words. For example, the work linked is used in several places, when I think the authors mean that the same serotypes occur I different populations. Also, the authors write that there are salmonella serotypes in wild foxes “responsible for infections in humans”. To my knowledge, it has not been shown that humans are infected by wild foxes, and I think the authors mean that the same serotypes are found in the different populations. The authors also write that the severity of salmonella infections in farmed foxes has been studied, when the study concerns occurrence of salmonella in two farms. The word determined is used, when I think the authors mean testing for associations. Please check that the correct expressions are used throughout the paper.

Author’s response:

The relevant corrections have been made.

Comment 6:

In line 55 the authors write that “salmonella is most prevalent in farms where veterinary and sanitary regulations are not observed”. Please include the reference for this.

Author’s response:

This sentence has been removed because the references should be limited to 30 items in Short Communications.

Once again, the authors would like to thank the Reviewer for the valuable remarks.